# Genetic Deletion of DNAJB3 Using CRISPR-Cas9, Produced Discordant Phenotypes

**DOI:** 10.3390/genes14101857

**Published:** 2023-09-24

**Authors:** Shadi Nejat, Kalhara R. Menikdiwela, Aliyah Efotte, Shane Scoggin, Bolormaa Vandanmagsar, Paul J. Thornalley, Mohammed Dehbi, Naima Moustaid-Moussa

**Affiliations:** 1Department of Nutritional Sciences & Obesity Research Institute, Texas Tech University, Lubbock, TX 79409, USA; snejat@ttu.edu (S.N.); kalhara.menikdiwela@rutgers.edu (K.R.M.); aliyah.efotte@ttu.edu (A.E.); shane.scoggin@ttu.edu (S.S.); 2Pennington Biomedical Research Center, LSU System, Baton Rouge, LA 70808, USA; bolormaa.vandanmagsar@pbrc.edu; 3Qatar Biomedical Research Institute (QBRI), Hamad Bin Khalifa University, Doha P.O. Box 34110, Qatar; pjthornalley@gmail.com (P.J.T.); mdehbi61@gmail.com (M.D.)

**Keywords:** obesity, type 2 diabetes, heat shock proteins, DNAJB3, adipose tissue, CRISPR-Cas9, inflammation, insulin resistance, glucose homeostasis, ER stress

## Abstract

Several pathways and/or genes have been shown to be dysregulated in obesity-induced insulin resistance (IR) and type 2 diabetes (T2D). We previously showed, for the first time, impaired expression of DNAJB3 mRNA and protein in subjects with obesity, which was concomitant with increased metabolic stress. Restoring the normal expression of DNAJB3 attenuated metabolic stress and improved insulin signaling both in vivo and in vitro, suggesting a protective role of DNAJB3 against obesity and T2D. The precise underlying mechanisms remained, however, unclear. This study was designed to confirm the human studies in a mouse model of dietary obesity-induced insulin resistance, and, if validated, to understand the underlying mechanisms. We hypothesized that mice lacking DNAJB3 would be more prone to high-fat (HF)-diet-induced increase in body weight and body fat, inflammation, glucose intolerance and insulin resistance as compared with wild-type (WT) littermates. Three DNAJB3 knockout (KO) lines were generated (KO 30, 44 and 47), using CRISPR-Cas9. Male and female KO and WT mice were fed a HF diet (45% kcal fat) for 16 weeks. Body weight was measured biweekly, and a glucose tolerance test (GTT) and insulin tolerance test (ITT) were conducted at week 13 and 14, respectively. Body composition was determined monthly by nuclear magnetic resonance (NMR). Following euthanasia, white adipose tissue (WAT) and skeletal muscle were harvested for further analyses. Compared with WT mice, male and female KO 47 mice demonstrated higher body weight and fat mass. Similarly, KO 47 mice also showed a slower rate of glucose clearance in GTT that was consistent with decreased mRNA expression of the GLUT4 gene in WAT but not in the muscle. Both male and female KO 47 mice exhibited higher mRNA levels of the pro-inflammatory marker TNF-a in WAT only, whereas increased mRNA levels of MCP1 chemokine and the ER stress marker BiP/Grp78 were observed in male but not in female KO 47 mice. However, we did not observe the same changes in the other KO lines. Taken together, the phenotype of the DNAJB3 KO 47 mice was consistent with the metabolic changes and low levels of DNAJB3 reported in human subjects. These findings suggest that DNAJB3 may play an important role in metabolic functions and glucose homeostasis, which warrants further phenotyping and intervention studies in other KO 47 and other KO mice, as well as investigating this protein as a potential therapeutic target for obesity and T2D.

## 1. Introduction

Obesity and Type 2 Diabetes (T2D) are increasingly prevalent globally, resulting in several comorbid conditions and health problems and severe financial burden, with annual healthcare expenses amounting to USD 147 billion [1,2]. Specifically, obesity is a multifaceted complex disease that is linked to a range of concurrent chronic diseases, including T2D, cardiovascular diseases, certain cancers, and liver diseases [3,4,5]. These comorbidities lead to a reduced life expectancy—in most cases, by 20%—compared with individuals with a healthy body weight [6,7,8]. Currently, over 42% of the adult population in the United States suffers from obesity, and the obesity prevalence is expected to increase continuously along with T2D during the next couple of years [1,9,10].

Genetic predisposition and environmental factors such as a sedentary lifestyle are among the key contributing factors to the etiology of obesity. Chronic low-level inflammation and cellular stressors like endoplasmic reticulum (ER) and oxidative stress, in addition to the reduced effectiveness of the heat shock response (HSR), are among the key defining features of obesity and various metabolic disorders [11,12,13,14,15,16].

Heat shock proteins (HSPs) are a group of highly conserved proteins that are crucial for alleviating diverse metabolic stresses on the body and aiding in the preservation of protein balance [17,18]. HSPs are named according to their molecular weight, amino acid sequences and their functions. There are six major classes of HSPs (HSP27, HSP40, HSP60, HSP70, HSP90 and large HSP110 and HSP170) in mammalian cells, each with their own unique functions [19]. HSPs are overexpressed or produced when the body is exposed to different stressors such as heat, cold, UV light or other stress conditions [20]. HSPs possess a wide range of functions and characteristics, encompassing the detection of alterations in the intracellular environment, the restoration and removal of damaged proteins, anti-apoptotic effects and interaction with the inflammatory transcription factors, including nuclear factor kappa B (NF-κB) in metabolic diseases [19,21,22,23,24,25,26]. The anti-stress and anti-inflammatory properties of HSPs are two of their best-known functions, which link them to the prevention of T2D [23,24,25,27,28]. The HSP40 family that includes the DNAJB3 protein, is cochaperone for HSP70; and some of its main functions include regulating ATPase activity and recruiting substrate proteins of HSP70 to promote protein folding [20,29]. If a protein continues to fail to fold correctly, then HSP40 assists the body in ridding itself of the protein and avoiding compiled misfolded proteins that could potentially lead to chronic disease [30,31].

Understanding the intricate relationship between HSPs and their co-chaperone DNAJB3 is pivotal in unraveling the mechanisms underlying protein homeostasis and its implications in health and disease, including Type 2 Diabetes (T2D). In light of these critical functions, our study seeks to investigate how alterations in DNAJB3 function, such as its inactivation or deficiency, may contribute to T2D pathogenesis or prevention.

In this study, given previously reported low levels of DNAJB3 in individuals with obesity and T2D, we investigated whether the lack of DNAJB3 would increase body weight, fat mass and glucose intolerance in mice fed high-fat diets [22]. We hypothesized that an animal model lacking DNAJB3 would have increased body weight, body fat, inflammation and glucose intolerance when fed obesogenic high-fat diets. Our hypothesis was tested using knockout mice lacking this protein, which were generated using CRISPR-Cas9 technology. Our results indicate that the absence of DNAJB3 adversely affects adipose and muscle metabolism, glucose clearance and inflammation. This is the first report that DNAJB3 deficiency could play a potential role in the development of obesity and associated glucose intolerance and inflammation. Our findings suggest that DNAJB3 may constitute a potential therapeutic agent in the treatment of obesity and diabetes, which warrants further basic and clinical research.

## 2. Materials and Methods

### 2.1. Generation of HSP40/DNAJB3 Animal Models

DNAJB3 knockout mice were generated using the CRISPR Cas-9 technology at the Transgenic Core facility of the Pennington Biomedical Research Center (PBRC, Baton Rouge, LA, USA). Briefly, three pairs of optimal sgRNA oligos were designed by scanning the target sequence close to start codon with CRISPR Design (http://crispr.mit.edu/, accessed on 9 July 2017) and cloned into Cas9-expression vector pX459-v2 (Addgene); the target sequence was used for the Dnajb3 gene (Gene ID: MGI:1306822). The T7 promoter sequence was added to the sgRNA template by PCR amplification, and the sgRNA synthesized in vitro with the transcription HiScribe™ T7 RNA Kit (New England Biolabs, Ipswich, MA, USA). Among the three sgRNAs, one was identified with the best efficiency in vivo with STO cells and in vitro digestion with the Cas9 enzyme. sgRNA and Cas9 protein were diluted and mixed in IDTE buffer (IDT, Coralville, IA, USA) and incubated at 37 °C for 5 min. One picoliter of the mixture was then injected into the pronuclei of one-cell stage zygotes from C57BL/6 J (B6) mice (The Jackson Laboratory, Bar Harbor, ME, USA). Therefore, these DNAJB3 knockout mice had the B6 background and were subsequently maintained for breeding by crossing to B 6 mice.

### 2.2. Animal Studies and Dietary Interventions

All experiments were conducted in compliance with the NIH Guide for the Care and Use of Laboratory Animals and approved by the Pennington Biomedical Research Center Institutional Animal Care and Use Committee (IACUC) (Protocol #982, approved on 4 April 2017). Breeding pairs generated at PBRC were shipped to Texas Tech University (TTU) for breeding and experimentation, and subsequent animal studies done at TTU following approved IACUC protocols (Protocols # 16011-04 and # 19034-04, approved on 8 April 2016; and 8 April 2019, respectively). Upon weaning, mice were randomly assigned to high-fat diet (HFD composed of protein 20% kcal, fat 45% kcal, and carbohydrates 35% kcal from Research Diets Inc. #D12451, New Brunswick, NJ, USA). After 16 weeks, DNAJB3-KO and WT littermates were euthanized in a CO_2_ chamber, and tissues harvested, snap-frozen in liquid N2, and stored at −80 °C for further analyses (study design illustrated in Figure 1).

### 2.3. Body Weight and Body Composition

Mice were weighed biweekly, and body composition was determined by nuclear magnetic resonance (NMR) using the LF110 BCA analyzer Bruker NMR Minispec (Bruker Corporation, Billerica, MA, USA) once a month. Fat mass and fat-free mass were obtained following calibration with internal standards as recommended by the manufacturer.

### 2.4. Glucose and Insulin Tolerance Test

Glucose tolerance tests (GTT) were conducted after fasting mice for 4 h. Briefly, mice were injected intraperitoneally (i.p.) a 20% D-glucose solution, where each mouse received 40 mg of glucose or 4 g/kg of body weight. After a baseline blood glucose reading (tail vein collection) using glucometer Breeze 2 (Ascensia Diabetes Care, Parsippany, NJ, USA), the warmed solution of saline D-glucose was injected i.p. Subsequently, blood glucose measurements were serially taken at 20, 40 and 60 min after the injection from the same tail site. Insulin Tolerance Tests (ITT) were conducted in the fed state, where mice were injected with 0.04 U kg^−1^ insulin (Eli Lilly and Company, Indianapolis, IN, USA), following which glycemia was measured at 0, 15, 30, 45, 60 and 90 min using the same glucometer. The area under the curve was calculated using the trapezoidal method with values normalized to the lowest basal glucose value. Due to unavoidable circumstances, GTT and ITT were performed only in male mice.

### 2.5. RNA Extraction, cDNA Preparation and Real Time PCR-Gene Expression

White adipose and skeletal muscle tissues were homogenized in 1 mL QIAzol Lysis Reagent (Qiagen, Valencia, CA, USA). Next, total RNA was isolated form these tissues using the RNeasy lipid tissue kit (Qiagen, Valencia, CA, USA), followed by cDNA synthesis using iScriptTM Reverse Transcription Supermix (Bio-Rad Laboratories, Inc., Hercules, CA, USA). Gene expression levels associated with inflammation, glucose metabolism, autophagy and endoplasmic reticulum stress were determined by real-time quantitative polymerase chain reaction (RT-qPCR) with Sybr green master mix (BioRad, Hercules, CA, USA). Data were normalized to the housekeeping gene, TATA Binding Protein (TBP). All primers were optimized prior to gene expression. The primer list is provided in Table 1.

### 2.6. Western Blot to Confirm DNAJB3

To validate absence of DNAJB43 in our KO mice, we extracted protein by lysing testes in modified RIPA buffer. Proteins were then separated by electrophoresis using gradient gels (Bio-Rad, Hercules, CA, USA), then transferred to polyvinylidene fluoride membranes (Sigma-Aldrich, St. louis, MO, USA). Next, membranes were kept in blocking buffer for 1 h at room temperature and then incubated with primary antibodies. The primary antibodies used in this study were raised against DNAJB3 (Proteintech Group, Inc., Chicago, IL, USA). After respective secondary antibody (rabbit polyclonal and mouse polyclonal, 1:20,000) incubations, the blots were imaged using the LI-COR Odyssey Imaging System (Lincoln, NE, USA) (Figure 2).

### 2.7. Statistical Analyses

Results are presented as means ± S.E.M. Student’s *t*-test was used with a *p* < 0.05 to compare each of the KO lines to WT littermates. Three to five replicates were used for each dietary or treatment group. All statistical analysis was conducted using GraphPad Prism Version 9.3.1 software.

## 3. Results

### 3.1. Genotyping

#### 3.1.1. Genotype Confirmation by Sequencing

The pups were genotyped with PCR using tail DNA. There were 62 pups born and a dozen of them were positive by PCR (Figure 2A). PCR products were sequenced to confirm the deletions in the targeted gene (Figure 2B). Three F0 lines (#30, 44, 47) were bred with wild-type C57BL6 mice for successful germline transmission. KO 30 had 40 bp deletion and KO 44 had 45 bp deletion that resulted in shift mutations that led to the truncated protein with different amino acid sequences in the end (Figure 2C), whereas KO 47 had 62 bp deletion that resulted in a loss of start codon ATG; consequently, no DNAJB3 protein was produced. We also sequenced tissue genomic DNA from each of these mice to validate the deletion.

#### 3.1.2. Deletion Confirmation by Western Blot

To validate the absence of DNAJB3, samples from testes of the three KO strains were analyzed by western blots, using DNAJB3 antibodies. Western blot results demonstrated that tubulin protein bands for all 3 lines were visualized at 52 kDa (Figure 3A); however, the DNAJB3, with a molecular weight around 37 kDa, was only seen in the WT group and in none of the KO strains (Figure 3B).

### 3.2. Effects of DNJAB3 Inactivation on Body Weight and Adiposity

To investigate how DNAJB3 deficiency influences body weight and fat mass gain, measurements were taken over a 16-week study period for both males and females across four different groups (WT, KO 30, KO 44, KO 47). Compared with WT mice, male KO 47 mice gained significantly more weight (*p* = 0.0400), which was observed at weeks 7, 11 and 16 of the study (Figure 4A,B). A similar increase in body weight was shown in female KO 47 mice, although more pronounced than that in male KO 47 mice (*p* = 0.0314) (Figure 4E). Moreover, the fat mass of both male and female KO 47 mice was significantly higher (*p* < 0.05) compared with their WT littermates, which was noted during weeks 7, 11 and 16 of the study (Figure 4B,F). By contrast, male and female KO 30 and 44 mice did not follow a similar trend, with KO 30 mice weighing less and KO 44 mice showing no changes in body weight, compared with WT mice (Figure 4A,E). Fat pad weights were significantly higher in KO 47 males and females compared with their WT counterparts (*p* = 0.0185 and *p* = 0.0074, respectively) (Figure 4D, H). Interestingly, KO 30 male mice showed significantly (*p* < 0.05) reduced fat pad weight compared with WT mice (Figure 4D). However, no significant changes were observed in lean mass/fat free mass across the three KO lines and WT in male or female mice (Figure 4C,G) further indicating a potential role of DNAJB3 in specifically regulating adiposity and fat metabolism.

### 3.3. Effects of DNAJB3 Deficiency on Glucose Homeostasis

As expected, KO 47 mice had significantly reduced glucose clearance at the 40- and 60-min time points compared with control mice (*p* < 0.05) (Figure 5A). However, male mice of the KO 30 and 44 lines did not follow the same trend and had a faster rate of glucose clearance rate compared with WT mice. The insulin tolerance test also followed a similar trend: the male mice of KO 47 demonstrated a slower rate of glucose clearance after insulin injection compared with WT mice (Figure 5C). This was also confirmed by area under the curve (AUC) values for both GTT and ITT (Figure 5B,D), which were different, compared with WT mice, for KO47 mice (*p* = 0.2312 and *p* = 0.4196, respectively). As obesity and glucose intolerance are linked to changes in glucose-metabolizing and insulin-signaling genes, we measured changes in some representative markers in the WT and three KO lines. In male white adipose tissue (WAT), the mRNA level for Glucose Transporter Type 4 (GLUT4) was significantly lower in KO 47 (*p* value <0.05) with no significant changes in WAT from KO 30 and KO 44; additionally, no changes were observed in female KO mice compared to WT mice (Figure 6A,C). Moreover, we did not observe any changes across all groups for the mRNA levels of the Insulin Receptor (INSR) in both the WAT and muscle tissues (Figure 6 and Appendix A). By contrast, we did not observe any differences in GLUT4 or INSR expression in muscle from males or females of the WT and three KO lines (Appendix A).

### 3.4. Effects of DNAJB3 Inactivation on Fat Metabolizing Genes

Given observed differences in body weight and fat mass, we further investigated the effects of DNJAB3 on lipid-metabolizing genes in WAT. The expression of lipogenic genes acetyl-CoA carboxylase α (Acaca), and fatty acid synthase (Fasn) was not different between KO 47 and WT mice for both sexes (Figure 7A,B,E,F). However, Acaca expression was significantly higher in KO 30 males, but not in females, compared with control WT mice (Figure 7A,E). On the other hand, KO 44 mice demonstrated a significantly lower Acaca expression in females, but not in males, when compared with WT littermates (Figure 7A,E). For fat-oxidation-related markers, such as carnitine palmitoyltransferase 1A (Cpt1a) and peroxisome-proliferator-activated receptor α (Ppara), male KO 30 mice had a significantly lower expression of Cpt1a compared with the WT and other KO lines; additionally, the female KO 47 mice showed a significantly higher expression of Cpt1a compared with the WT and other KO lines. Moreover, Ppara was trending higher in KO 30 males and KO 47 females (Figure 7C,D,G,H).

### 3.5. Effects of DNAJB3 Deficiency on WAT and Muscle Inflammatory Markers

Given the association between body weight and fat gain, and the interconnection of glucose and insulin intolerance with chronic low-grade inflammation, we assessed alterations in expression levels of inflammatory markers in both adipose and muscle tissues. Expression of tumor necrosis factor α (Tnfa) was significantly upregulated in WAT from male and female KO 47 mice (*p* = 0.0278 and *p* = 0.0045, respectively) compared with control WT littermates (Figure 8A,D). By contrast, no changes were observed in Tnfa expression for male or female KO 44 mice when compared to WT littermates. Interestingly, KO 30 mice exhibited a significantly lower WAT Tnfa mRNA level in males compared with WT littermates (Figure 8A). Similarly to Tnfa expression in male WAT, monocyte chemoattractant protein1 (Mcp1) levels were substantially increased in KO 47 mice (*p* = 0.0178) with no significant changes in male or female KO 30 and 44, when compared to control WT littermates (Figure 8B). Again, no changes in Mcp1 expression were seen in muscles from any KO strain compared with WT males or females (Appendix A). Other genes related to inflammation, including Interleukin 6 (IL6), were also tested; in white adipose tissue of male KO 47 mice, compared to the control group, we did not observe any differences. In female WAT from KO 47, the IL6 mRNA level was trending higher (*p* = 0.0540) compared to that of WT littermates (Figure 8C,F). In muscle tissues, mRNA expression of IL6 was not significantly different between KO and WT littermates in male mice; there was an upward trend for KO 47 compared to WT (*p* = 0.4748). However, in females, muscle IL6 mRNA level was significantly higher in KO 44 (*p* = 0.0168) and KO 47 (*p* = 0.0476), compared with WT littermates (Appendix A).

### 3.6. Effects of DNAJB3 on ER Stress Markers

Overexpansion of adipocytes due to fat accumulation in obesity, causes cell stress, including endoplasmic reticulum (ER) stress [32]. Thus, to understand whether a lack of DNAJB3 induces ER stress, we tested several ER stress markers such as Binding Immunoglobulin Protein (BiP), Activating Transcription Factor 6 (ATF6), and C/EBP Homologous Protein (CHOP). Although in the male WAT tissue we observed trends towards upregulation of ATF6 (*p* = 0.0629) in the KO 47 group compared with the WT group, there were no changes in KO 30 or 44 (Appendix A) or in female mice or in muscle tissues (Appendix A). We did observe a significantly higher expression of BiP in male WAT of the KO 47 line; however, we did not observe any differences in CHOP and BiP expression when comparing the other KO groups to WT in adipose or muscle tissues (Appendix A).

## 4. Discussion

In this current study, we evaluated the effect of 16 weeks of high-fat (HF)-diet-induced obesity on body weight, metabolic stress and glucose metabolism in an experimental mouse model lacking DNAJB3. In response to the HF diet, DNAJB3 KO animals exhibited increased body weight with concomitant adverse effect on blood glucose clearance and increased inflammatory and ER stress responses in WAT as compared with their wild-type littermates. Consistent with our previous in vivo and in vitro findings [17,22], the results of this investigation suggest protective roles for DNAB3 in preventing weight gain, triggered by an HF diet and in mitigating the well documented metabolic alterations in obesity, that lead to loss of glucose homeostasis [5].

DNAJB3, which is also known as Msj-1 in mice, is a member of the HSP40 family; it acts as a cochaperone to HSP70 by activating ATPase activity to ensure the correct folding and remodeling of proteins in the body—a prerequisite condition to maintaining optimal cellular function and preserving tissue integrity [19,26,33,34]. The primary function of HSPs in maintaining normal protein homeostasis or proteostasis and preventing the accumulation of misfolded proteins in the body makes them a great target to prevent several chronic metabolic and neurological diseases [17,22,35,36,37,38,39]. Besides their role in acting as cellular guardians of the proteostasis network, recent studies documented a clear role of HSPs in binding and/or controlling the activity of key enzymes and pathways involved in inflammation, apoptosis, metabolism and cell signaling [17,27,28,31]. Genetic manipulation of certain HSPs or modulation of their expression provided key evidence for their role in the pathogenesis of a wide range of chronic diseases such as neurological disorders, non-alcoholic fatty liver disease (NAFLD) and diabetes [17,22,24,25,38,39,40].

We previously reported impaired expression of DNAJB3 mRNA and protein in the WAT of subjects with obesity, that was associated with increased inflammatory response and altered metabolic profile [22]. The same study demonstrated the effectiveness of a 6-month physical exercise protocol in restoring the normal expression of DNAJB3 in those subjects, with significant improvement in body fat content and cardiopulmonary performance, and reduced inflammatory and stress responses [22,41]. Subsequent in vitro studies confirmed a direct effect of DNAJB3 overexpression and pharmacological modulation of its expression in promoting glucose metabolism and insulin signaling while mitigating various forms of metabolic stress, namely ER stress as well as the κB- and JNK-dependent pathways [17,35,41]. Corroborating those findings, our data demonstrated that a deficiency of DNAJB3 increased body weight and fat mass in KO 47 mice, further indicating a potential role of this target in regulating body weight, fat mass and glucose homeostasis.

Our results revealed that the absence of DNAJB3 altered glucose metabolism significantly by reducing the glucose-clearing ability in mice. A similar correlation between HSP levels and glucose metabolism was reported by others [40,42,43]. Reduced expression of HSP72 was evident in streptozotocin-induced diabetic rats, and reduced expression of both HSP25 and HSP72 was seen in insulin-resistant aged rats [40,42]. Additionally, mice lacking HSP72 were shown to be phenotypically obese, displayed glucose intolerance and IR in skeletal muscle [40]. These findings suggest that HSPs including DNAJB3 play a role in regulating glucose metabolism. Thus, it is plausible that ablation of DNAJB3 affects the insulin-signaling pathway in metabolically relevant tissues, including adipose tissue, causing impairments in systemic glucose clearance. Reduced levels of mRNA levels for insulin-signaling markers (e.g., Glut4) were observed in KO47 males in adipose tissue. We did not, however, observe the same trend in our other KO lines (30 and 44). To further investigate the differences observed among the three knockout lines, additional animal studies, including a total DNAJB3 knockout mouse model, generated using other approaches are warranted.

Obesity is linked to persistent chronic low-grade inflammation, which is characterized by the improper expression and release of various pro-inflammatory cytokines/chemokines within adipose tissue, along with adipose tissue remodeling and sustained ER stress and oxidative stress [32,44]. HSPs play a protective role against obesity-induced, insulin resistance, in part by virtue of their anti-inflammatory and anti-stress properties [17,35,40,41,45,46]. HSPs form physical protein complexes to suppress the activity JNK and IKKß, two crucial enzymes that interfere with the insulin signaling [17,38,39,41]. Therefore, any alteration in the expression levels of HSPs has profound effects on the activation and/or release an array of inflammatory and stress markers that culminate in a loss of glucose homeostasis [28,45,46].

Based on our previous findings, we postulated that DNAJB3 KO mice would have increased inflammatory and ER stress and altered glucose metabolism in response to a high-fat diet as compared with the wild-type group. As expected, mRNA levels of representative of the inflammatory markers (MCP1, TNFa and IL-6), and ER stress markers (ATF 6, CHOP and BIP/GRP78), were altered mostly white adipose tissue from male and female KO47 mice. To our surprise, we did not observe such changes in the skeletal muscle tissue of these KO animals. This could be in part due to the limited number of samples we had available in each group, which warrants more powered and extensive studies.

While the current study sheds new light on the role of HSPs in obesity and its related metabolic functions, there are a few limitations. We lack low-fat diet data in this study as well as glucose tolerance test data for female mice. We observed different results in our three mutant lines, which will be further investigated in more extensive studies with a higher number of replicates to understand the basis for these discrepancies. There were no prior animal studies that had inactivated this gene, and the very limited published work on the role of DNAJB3 further complicates our understanding the functions of this gene in metabolic diseases.

Our data demonstrated that deficiency in DNAJB3 increased body weight and fat mass in one of our knockout lines. The reason for the difference in the three lines could possibly be in part due to a discrepancy in the CRISPR-Cas 9 technology that was utilized to create these mutations. The mutation for line 47 began at the start codon and there is a possibility that the deletion of the gene at the start codon is responsible for the obesity and metabolic changes that were observed in that line.

In conclusion, the absence of DNAJB3 potentially increases adiposity, glucose intolerance and inflammation in high-fat-diet-induced obese mice. DNAJB3 potentially plays a major role in metabolic functions and glucose metabolism, which warrants further research on it as a potential therapeutic agent. Given the growing number of people suffering from metabolic diseases, our research could potentially pave new ways to battle these non-communicable chronic diseases.

## Figures and Tables

**Figure 1 genes-14-01857-f001:**
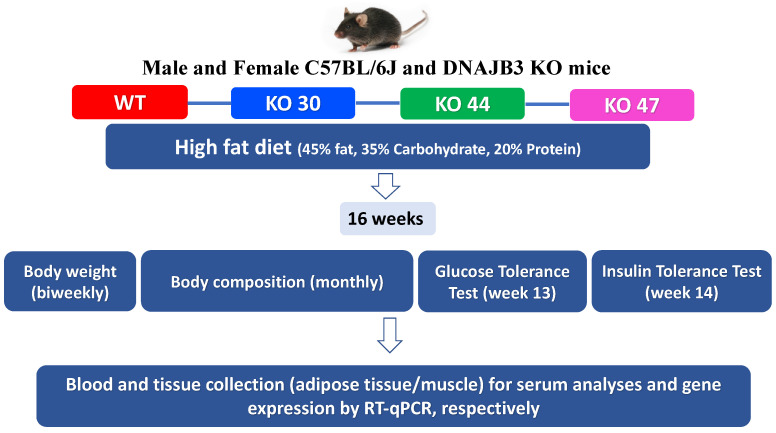
Study design.

**Figure 2 genes-14-01857-f002:**
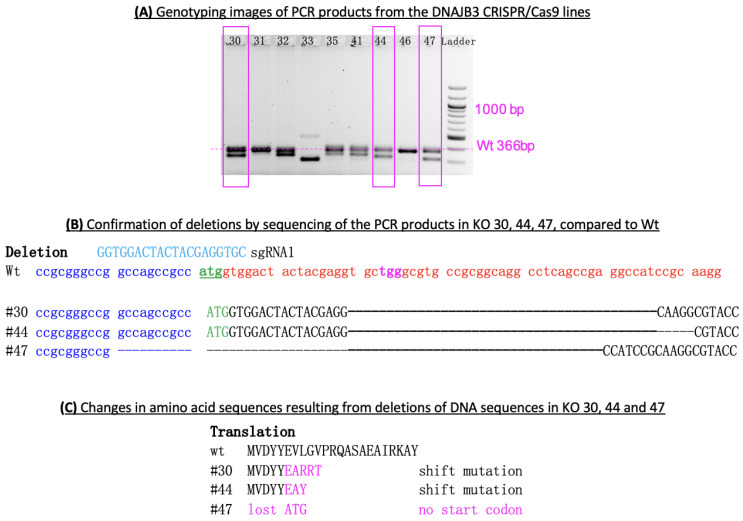
Genotyping strategy. (**A**) Gel image of genotyping PCR products: the band with 366 bp indicates the Wild type (Wt) genotype, while bands shorter than 366 bp show deletions in the targeted region of the DNAjb3 gene, thus indicating a knockout genotype. The 100 bp DNA ladder (New England BioLabs, Ipswich, MA) was used in the gel image (1000 bp band in the ladder is marked). Numbers on top of the lines indicate strain/KO number, of which we focused on KO 30, 44 and 47. (**B**) The deletions were confirmed by sequencing of the PCR products in three KO lines #30, 44, 47: targeted sequences are marked by red; deleted sequences within the targeted region are underlined and italic; the initiation code ATG is marked by green; and a binding signal for Cas9 nuclease (TGG as a PAM sequence) is marked by light blue. (**C**) The deletions in DNA sequences (upper rows) resulted in changes in amino acid sequences (lower rows) in three KO lines: the sequence of sgRNA is shown in light blue; the target sequence in DNAJB3 in WT genotype is shown in red; and deleted sequences in all three lines are marked by dotted lines.

**Figure 3 genes-14-01857-f003:**
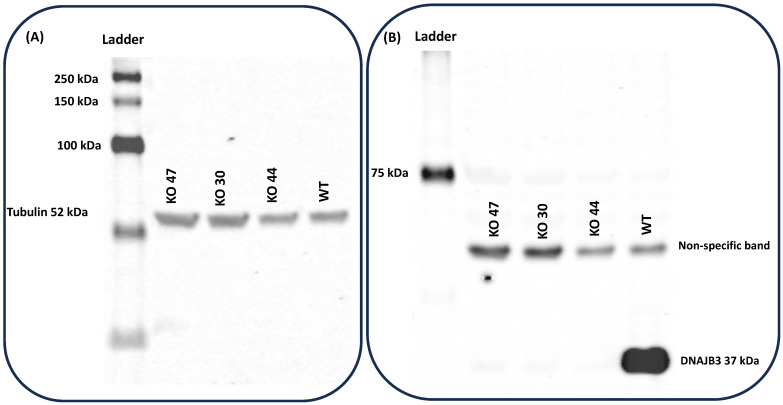
Validation of absence of DNAJB3 in protein samples from testes of knockout strains. (**A**) Western results demonstrated control protein bands for Tubulin at (52 kDa) for all 3 lines including 30, 44, 47 and WT. (**B**) DNAJB3 band was only seen in WT lane but not in KOs.

**Figure 4 genes-14-01857-f004:**
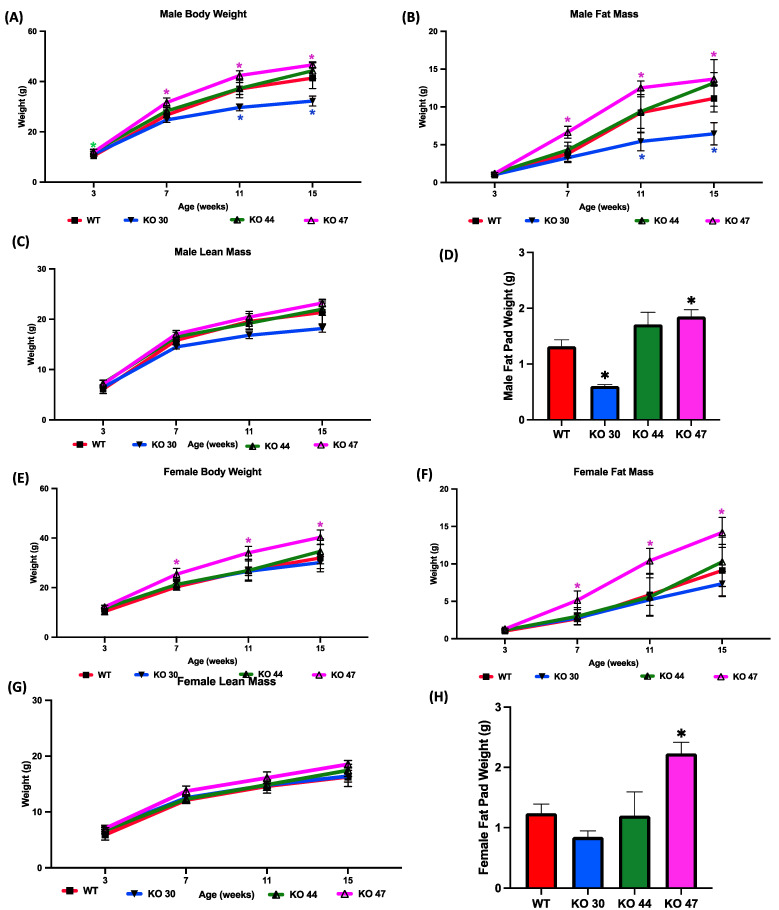
Effects of DNAJB3 inactivation on body weight and fat mass in male and female mice. Comparison of male and female KO vs. WT mice: (**A**,**E**) body weight, (**B**,**F**) fat mass, (**C**,**G**) lean mass, (**D**,**H**) fat pad weight. An asterisk indicates statistical significance, compared to respective WT time points (**A**,**B**,**E**,**F**) or to the WT group (**D**,**H**). Data are presented as mean ± SEM (n = 6). *p* < 0.05.

**Figure 5 genes-14-01857-f005:**
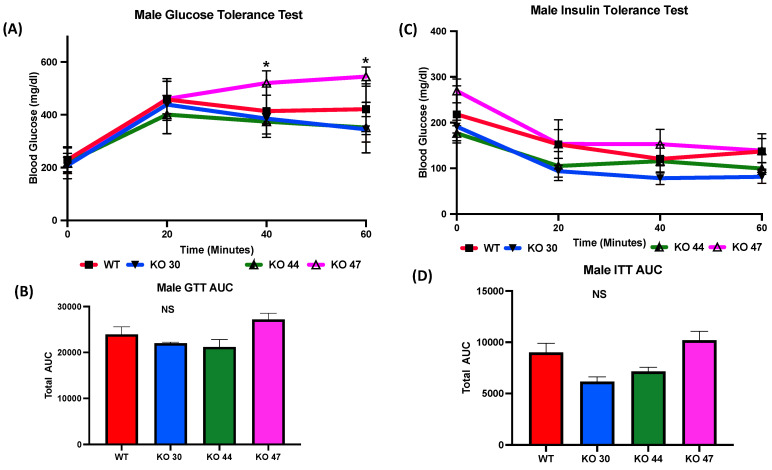
Role of DNAJB3 on glucose clearance in diet induced obese male mice. (**A**) Glucose tolerance test (GTT) in male mice. (**B**) Area under the curve (AUC) for GTT in male mice. (**C**) Insulin tolerance test (ITT) for male mice. (**D**) AUC for ITT male mice. An asterisk indicates statistical significance, compared to WT, while NS indicates no statistical significance across groups or compared to WT. Data are presented as mean ± SEM (n = 6). *p* < 0.05.

**Figure 6 genes-14-01857-f006:**
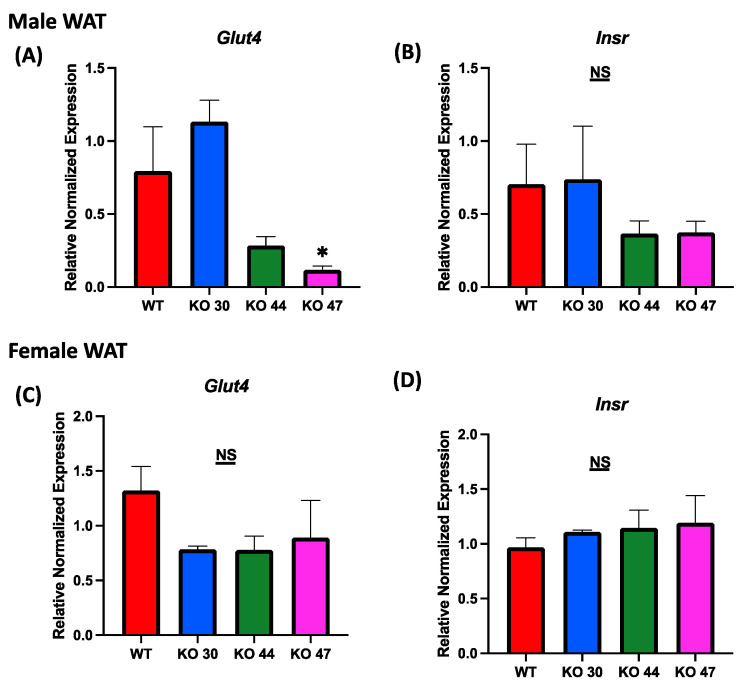
Role of DNAJB3 on regulating glucose metabolism in the adipose tissue of diet induced obese male and female mice. (**A**,**C**) mRNA levels of Glucose Transporter Type 4 (GLUT4) in white adipose tissue of male and female mice. (**B**,**D**) mRNA levels of Insulin Receptor (INSR) in white adipose tissue of male and female mice. An asterisk indicates significance, compared to WT, while NS indicates no statistical significance across groups or compared to WT. Data are presented as mean ± SEM (n = 6). *p* < 0.05.

**Figure 7 genes-14-01857-f007:**
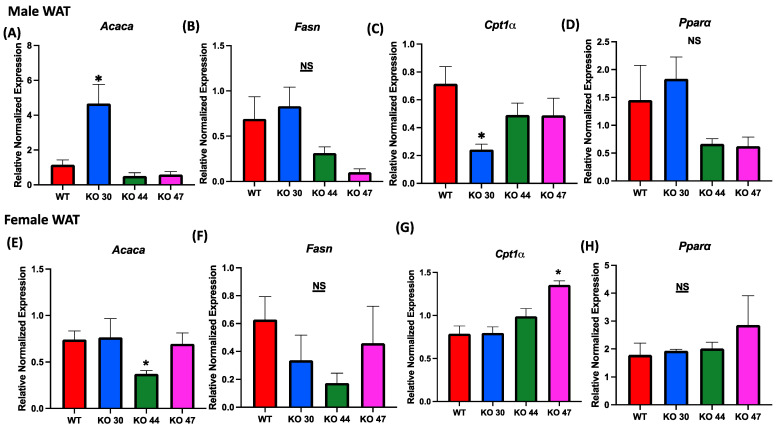
Role of DNAJB3 on fatty acid metabolism in white adipose tissue. (**A**,**E**) mRNA levels of acetyl-CoA carboxylase α (Acaca) in white adipose tissue of male and female mice. (**B**,**F**) mRNA levels of fatty acid synthase (Fasn) in white adipose tissue of male and female mice. (**C**,**G**) mRNA levels of carnitine palmitoyl transferase 1A (Cpt1a) in white adipose tissue of male and female mice. (**D**,**H**) mRNA levels of peroxisome-proliferator-activated receptor α (Ppara) in white adipose tissue of male and female mice. An asterisk indicates significance, compared to WT, while NS indicates no statistical significance across groups or compared to WT. Data are presented as mean ± SEM (n = 6). *p* < 0.05.

**Figure 8 genes-14-01857-f008:**
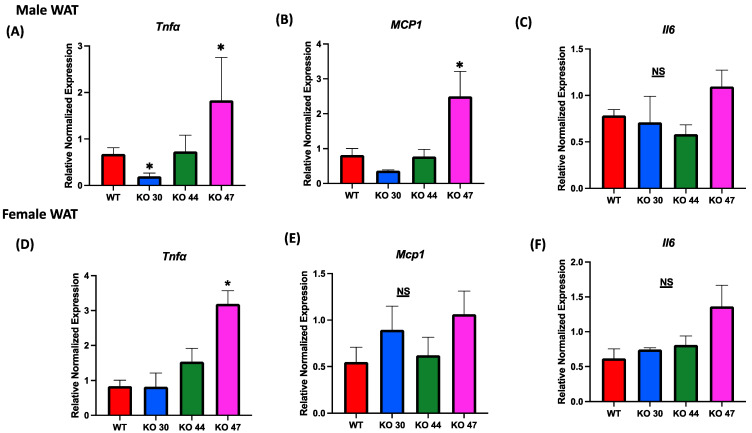
Role of DNAJB3 on inflammation in diet induced obese male and female mice. (**A**,**D**) mRNA levels of tumor necrosis factor a (Tnfa) in white adipose tissue of male and female mice. (**B**,**E**) mRNA levels of monocyte chemoattractant protein1 (Mcp1) in white adipose tissue of male and female mice. (**C**,**F**) mRNA levels of Interleukin 6 (IL6) in white adipose tissue of male and female mice. An asterisk indicates significance, compared to WT, while NS indicates no statistical significance across groups or compared to WT. Data are presented as mean ± SEM (n = 6). *p* < 0.05.

**Table 1 genes-14-01857-t001:** Primer list and sequences.

Gene	Forward	Reverse
*Acaca*	GCAGCAGTTACACCACATACA	CATTACCTCAATCTCAGCATAGCA
*Cpt1α*	GAGACAGACACCATCCAACAC	GAGCCAGACCTTGAAGTAACG
*Fasn*	TGCAGAAGATGTAGATTGTGTGATGA	GGGTCCGGGTGCAGTTTATT
*Pparα*	ATCCACGAAGCCTACCTGAA	AATCGGACCTCTGCCTCTT
*Glut 4*	AGAGCGTCCAATGTCCTT	CGAAGATGCTGGTTGAATAGTAG
*Insr*	ACCTTCCAGTATGTTCCTCAG	TGCCTTCAGTCATTACCTCTT
*Il6*	AACCGCTATGAAGTTCCTCTC	TCCTCTGTGAAGTCTCCTCTC
*Mcp1*	ACTTCTATGCCTCCTGCTCAT	GCTGCTTGTGATTCTCCTGTAG
*Tnfα*	CGTGGAACTGGCAGAAGAG	TGAGAAGAGGCTGAGACATAGG
*Atf 6*	TTCCTCCAGTTGCTCCATCT	ACCAGTGACAGGCTTCTCTT
*Bip*	TTCAGCCAATTATCAGCAAACTCT	TTTTCTGATGTATCCTCTTCACCAGT
*Chop*	CCACCACACCTGAAAGCAGAA	AGGTGAAAGGCAGGGACTCA

## Data Availability

All of the data supporting this work will be made available from the corresponding author upon reasonable request.

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
