# Peer review of "Genetic Deletion of DNAJB3 Using CRISPR-Cas9, Produced Discordant Phenotypes"

_genes, 2023, doi:10.3390/genes14101857_

Round 1
Reviewer 1 Report
-The article offers a comprehensive overview of the research conducted to explore the role of DNAJB3 in obesity and diabetes. The introduction sets out a complete scenario highlighting the prevalence and impact of obesity and type 2 diabetes, as well as the action of heat shock proteins in terms of regulating different mechanisms in the body and how this would impact these pathologies. It is an adequate context of the DNAJB3 to understand the scope of the study.
-It is well structured, it allows to correlate the molecular with the phenotypic findings and the clinical implications. The methodology is clear despite covering very specific information, it establishes a good framework to understand the generation of animal models, the dietary interventions and the various measurements taken during the experiment. The inclusion of specific details about the procedures improves the clarity of the study methodology.
-The use of CRISPR-Cas9 technology to investigate the effects of DNAJB3 deletion on phenotypes related to obesity and glucose intolerance is an innovative approach, I find it an interesting article to emphasize the use of technologies to the medical area.
-The results section effectively summarizes the results of the study. It highlights the effects of DNAJB3 deficiency on body weight, adiposity, glucose homeostasis, inflammatory markers, and endoplasmic reticulum stress markers. The use of tables and figures helps to present the data in a visual way, which makes it easier to interpret the findings, which in my opinion is of great importance considering the complexity of the information that is being processed.
-However, some sections of the article could be improved in terms of clarity and precision. In the introduction, a clearer connection between heat shock proteins and the role of DNAJB3 as a co-chaperone could be established. In addition, the description of the methodology could benefit from more detail on the rationale for choosing specific methods and controls, also taking into account the fact that there were three groups of animal models and only one of which saw significant results.
Author Response
Responses attached. Thank you

Reviewer 2 Report
The authors should improve on the following:
1. The abstract should be more concise and clearly depict the entire paper. It should not contain numbered subsections.
2. In Figure 2A, the entire ladder should be marked, not just the 366 bp.
3. Section 3.1.2., the procedure for western blots should be included in Methods, not in Results.
4. Figure 3, I am not sure how the authors obtained both Tubulin and DNAJB3 from the same piece of membrane. Was the incubation kept with two primary antibodies together? What's the topmost band? The ladder on the left has to be marked, otherwise, it can't be confirmed if the bands correspond to what is marked.
Please provide the original western blot image in better quality. The quality of the image provided at present is not upto the mark. Why are there colours on the blot image?
5. Figures 2,3 and the respective subsections can just be made into one figure, the conclusion is the same.
6. Figures 4,5 and the respective subsections can be combined into one figure. The data for male and female mice can be shown side by side on the same graph for better visualization. The graphs should be improved so that the lines can be seen properly.
7. Are there data on glucose and insulin tolerance tests for female obese mice?
8. Do the authors have any representative images of the isolated WAT?
Apart from all these, the authors should go through the entire paper, and make all the sections more concise. There should be appropriate conclusions after each Results section, not just writing down what can be seen in the graphs.
There are references missing in many sections. For example, "In this study, given previously reported low levels of DNAJB3 in individuals with obesity and T2D," (lines 77-78) reference is missing.
Catalogue numbers should be mentioned for all the obtained chemicals, and antibodies.
English improvements are necessary.
Author Response
Responses attached thank you

Round 2
Reviewer 2 Report
The paper can be accepted in its present form